# Anticipatory Awareness and Actual Handwriting Performance Measures among Adolescents with Deficient Executive Functions

**DOI:** 10.3390/children9111628

**Published:** 2022-10-26

**Authors:** Yael Fogel, Sara Rosenblum

**Affiliations:** 1The Laboratory for Promoting Daily Executive Functioning, Department of Occupational Therapy, Faculty of Health Sciences, Ariel University, Ariel 40700, Israel; 2The Laboratory of Complex Human Activity and Participation (CHAP), Department of Occupational Therapy, Faculty of Welfare and Health Sciences, University of Haifa, Haifa 3498838, Israel

**Keywords:** adolescent, handwriting, executive function, anticipatory awareness, time estimation, time duration, legibility

## Abstract

This secondary analysis explores differences and correlations between handwriting anticipatory awareness (time estimation, expected performance, and expected difficulty) before a paragraph copying task and actual performance time and legibility among adolescents with executive function deficits (EFD) versus neurotypical adolescents. Eighty-one adolescents (10–18 years old; 41 with EFD and daily functioning difficulties as per parent reports) estimated their time, expected performance, and expected difficulty before the paragraph copying task using the Self-Awareness of Performance Questionnaire (SAP-Q). Time duration was assessed using the Computerized Penmanship Evaluation Tool (ComPET) software, and legibility was scored using the Handwriting Legibility Scale (HLS). Significant between-group differences were found in actual time duration (ComPET), HLS total score and legibility components, and three SAP-Q questions. Both groups estimated significantly more performance time than their actual performance duration. The adolescents with EFD underestimated their performance before the handwriting task. Significant correlations were found between actual performance, anticipatory awareness, and Executive Function (EF) in both groups. Their performance predictions were significantly correlated with their EF and product’s legibility. The results highlight the potential relationships between anticipatory awareness and actual handwriting performance (time duration and legibility) of adolescents with and without EFD. Further studies may analyze the benefits of focusing on both EF and anticipatory awareness for handwriting improvement among populations with EF deficits.

## 1. Introduction

Handwriting is an essential activity in which school-aged children engage [1]. Studies have shown the importance of fluent handwriting, highlighting positive relationships between the quality of written work, text length, and handwriting speed for children, including those with literacy, language, or motor difficulties [2,3]. Teachers frequently set copying tasks for students in the classroom, for example, copying rules or sentences from the board and writing homework instructions [4]. When copying these words or letters, the children must control the force and fine movement of their hands and fingers, pay attention to relevant stimuli, coordinate their visual with their manual movements, keep the target text to be copied in their working memory, and incorporate the orthographic structure’s linguistic rules with this motor and perceptual information. Such coordination of all these elements when copying words and letters helps ensure automatic, legible, and fluent handwriting as crucial indicators of skill mastery [5]. The research on handwriting has been aimed at evaluating handwriting through process and product measures [6].

Most handwriting processes and activities such as copying script require the writer’s focused attention, supported by intensive use of brain resources [7]. In the literature, it has been reported that the many processes involved in handwriting skills depend on executive functions (EF) [8,9]. These EF components, such as working memory [10,11], inhibition and shifting [12], planning, monitoring [13], decision making, and control of attention [14], help regulate handwriting efficiency [9,15]. Ineffective or inconsistent EF use—or disuse—may lead to challenges in text production, speed, formation, and automaticity [16,17].

Adolescents with EF deficit (EFD) characteristics have trouble planning, monitoring, and integrating the cognitive and motor components needed to produce handwriting efficiently [18,19]. The mechanism underlying these challenges and impairing daily functions, including handwriting performance [20,21,22], may be deficient EF. It has been found that children and adolescents with different neurodevelopmental difficulties with EF differ from their peers in handwriting process and product measures [1,11,16,18,23].

Handwriting for adolescents with EFD is challenging; it requires substantial effort. It is well-known that children and adolescents who struggle with handwriting activities may give up after repeated failures and believe they are incapable of improving their skills [6]. Avoiding writing can lead to negative psychosocial and academic consequences, such as reduced self-esteem, being mislabeled as lazy, and poorer grades [24,25,26]. In this context, awareness of their actual performance is important.

Self-awareness is a complex, multidimensional construct that includes distinct but interrelated concepts [27,28]. Recent years have seen more studies on self-awareness of performance among children and adolescents [29,30,31]. However, few studies have examined self-awareness of handwriting performance and its relationships with actual performance [32]. Lahav et al. [32] are one of the few examples. They found gender differences concerning neurotypical adolescents’ self-awareness of their handwriting performance. The self-awareness questions assessed participants’ overall handwriting performance beyond the context of a particular handwriting task (self-knowledge awareness) and just after performing the task (online awareness).

This study focuses on anticipatory awareness, the highest awareness level in Crosson et al.’s [33] model. *Anticipatory awareness* is a person’s ability to expect a problem due to some deficit and to act to prevent its occurrence. In Toglia and Kirk’s [27] awareness model, anticipatory awareness reflects good metacognitive knowledge and online awareness manifested in behavior. As far as we know, no previous study has explored anticipatory awareness related to handwriting performance. This study analyzes anticipatory awareness related to the handwriting performance time and actual performance (legibility) of adolescents with deficient EF profiles (evaluated by standardized assessments such as the Behavior Rating Inventory of Executive Function (BRIEF)) [34].

Assessing anticipatory awareness is crucial because, within the context of an activity, a person may demonstrate poor online awareness but good general awareness (or vice versa) [28]. It has been hypothesized that poor awareness of their challenges bars adolescents with attention deficit hyperactivity disorder (ADHD) and EFD from engaging and persisting in treatment [29,35,36]. However, not enough is known about how best to assess this construct in general or in the pediatric population specifically [37].

Researchers have reported that individuals with attention disorders, often characterized as having EFD, may exhibit positive illusory self-bias. Their self-ratings tend to overestimate their functioning in social and academic situations [38] and their actual performance [39,40]. Some studies have hypothesized that this bias might be a protective strategy, helping them persevere through challenges and overcome frequent setbacks [41]. Another explanation for this overestimation is that their awareness of their EF limitations is poor; hence, their assessment before performing specific tasks may be based on insufficient information about task complexity. These adolescents may have trouble systematically gathering information about the task, planning, and thinking through its challenges beforehand [42]. This difficulty may lead to a disassociation (gap) between their self-assessment before and actual performance during the task. Handwriting is a significant occupation and measure of academic achievement in adolescence. However, there is a lack of evidence on how adolescents’ self-awareness before performing a handwriting task (anticipatory self-assessment: time estimation, expected performance, and expected difficulty) can affect their actual handwriting-performance process (time duration) and product (legibility).

Thus, this study aimed to explore (1) actual handwriting performance (time duration and legibility) and anticipatory awareness of adolescents with and without deficient EF and (2) correlations between actual performance, anticipatory awareness, and EF components in each group. To answer the study’s aims, we analyzed the following components: (1) actual handwriting performance (time and legibility) between-group differences, (2) actual handwriting performance (time and legibility) correlations with EF subscale scores in each group, (3) handwriting anticipatory awareness time estimation between-group differences, differences between time estimation and time duration by group, and correlations between the time estimation/duration gap and EF subscale scores by group, and (4) handwriting anticipatory awareness expected performance and expected difficulty between-group and by-group differences, differences between expected performance and expected difficulty and legibility by group, and correlations between expected performance and difficulty and EF by group.

## 2. Materials and Methods

### 2.1. Participants

This study was a secondary analysis of a study that included 81 young adolescents aged 10 to 14 years with and without EFD profiles [34,43]. The participants were recruited by advertising in the community aimed at young adolescents with and without daily functioning challenges. The research group included 29 (70.7%) boys and 12 (29.3%) girls (age M = 11.88 years; SD = 1.08). The control group comprised 40 age- and gender-matched neurotypical adolescents: 28 (70%) boys and 12 (29.3%) girls (age M = 12.19 years; SD = 1.08). There were no significant between-group differences for demographic measures. We considered participants to have an EFD profile if their parent-reported scores on the BRIEF behavioral regulation or metacognition indices were above the normal range (65 or higher) [44]. We found significant differences (*p* = 0.000, η*p*2 = 0.43–0.74) between groups in the BRIEF global, index, and scale scores, with the study group having the lower mean [34,43]. We excluded participants with known physical disabilities; neurological diseases; emotional, psychiatric, or autism spectrum disorders; or taking psychotropic drugs.

### 2.2. Procedure

The Ethics Committee of the University of Haifa approved the study (253/13), and all adolescent participants and their parents signed informed consent forms. To address potential bias, one occupational therapist evaluated participants according to the inclusion and exclusion criteria. Another occupational therapist with relevant theoretical knowledge and experience with adolescents and handwriting scored the manuscripts using the Handwriting Legibility Scale (HLS). The raters were blinded to the participants’ EF profiles.

### 2.3. Instruments

#### 2.3.1. Behavior Rating Inventory of Executive Function

The BRIEF parent’s report [44] includes 86 questions divided into the metacognitive index (MI) and behavioral index (BRI). The MI includes scales for initiation, working memory, plan-organize, organization of materials, and monitoring. The three BRI scales are inhibit, shift, and emotional control. The global executive composite (GEC) is the BRIEF overall score. The results are expressed in t-scores (with lower scores reflecting less problematic behavior). A t-score of 65 or more is considered in the clinical range. The BRIEF parent version has good convergent and divergent validity, high internal consistency in the general population (78–0.96), and good test–retest stability (0.72–0.84).

#### 2.3.2. Handwriting Process: Computerized Penmanship Evaluation Tool (ComPET)

The ComPET (previously POET) [6] is a validated, standardized handwriting assessment that does not depend on language. Its use requires online data collection via a digitizing tablet and analysis software. A paragraph (46 words) in manuscript Hebrew font (Gutman Yad Brush) size 20 was placed in front of the participants [45]. Using a wireless electronic pen with a pressure-sensitive tip (Model GP-110), they copied the paragraph script onto A4-sized lined paper affixed to the surface of a WACOM Intuos II x-y digitizing tablet (404 × 306 × 10 mm). For the current analysis, time duration data (in s) were extracted from the ComPET because it best reflected actual performance.

#### 2.3.3. Handwriting Product: Handwriting Legibility Scale [46]

Developed originally using scripts from the Detailed Assessment of Speed of Handwriting writing task [47], the HLS [46] examines performance in five handwriting legibility components:legibility (readability)effort (required to read the script on the first attempt)letter formation (necessary elements, appropriate shapes, consistent size and slope, and neat letter closures)layout on the page (words organized on the page, within the margins, sufficient space between words and letters, and positioned on the baseline)alterations (additional strokes or retracing letters)

Each component is scored from 1 (good performance) to 5 (poor performance). Component scores add up to a total legibility score (5–25), with lower scores reflecting better legibility. Inter-rater reliability and internal consistency have been reported as high (0.92 each), with all components loading on just one factor [48,49]. In the current study, an occupational therapist with specific expertise in handwriting rated the HLS scores. Good reliability (0.80) was found in the current study.

#### 2.3.4. Anticipatory Self-Awareness of Performance Questionnaire

The clinician-administered Self-Awareness of Performance Questionnaire (SAP-Q) assesses the anticipatory awareness of performance [28], modified for the specific task examined. Before performing the handwriting task, participants were told that the task included copying script as they did in daily life. Then, the examiner asked the participants three questions related to estimated time (“How long do you think it will take you to perform the handwriting task?”), expected performance (“How do you think you will do on the handwriting task?” or “How legible do you think your product will be?”), and expected difficulty (“Do you think you will have difficulty performing this handwriting task, that is, writing legibly in a reasonable time?”). The expected performance question was rated 1 (bad), 2 (fairly good), 3 (good), 4 (very good), or 5 (excellent), and expected difficulty rated 1 (never), 2 (seldom), 3 (sometimes), 4 (often), or 5 (always).

### 2.4. Data Analyses

We processed the data using SPSS 26. Because the sample did not distribute normally, we used nonparametric tests. Mann–Whitney tests were conducted for differences between the groups in the process (actual time duration) and product (HLS) measures. Spearman’s correlation was used to analyze all correlations between handwriting processes and products and the EF in each group. Crosstab statistics and chi-square tests were used for the SAP-Q between-group differences and correlations, and the Wilcoxon test for within-group differences between estimated and actual duration.

Two groupings were made to identify relationships between expected performance and difficulty and total legibility score within each group. The HLS final (total) scores indicated the participant’s legibility group: 5–10 (excellent legibility), 11–15 (good legibility), or 16–25 (poor legibility). The SAP-Q scores were recorded in three groups of expected performance (bad + fairly good, good, and very good + excellent) and expected difficulty scores (always + often, sometimes, and seldom + never).

## 3. Results

### 3.1. Actual Handwriting Performance (Time and Legibility): Between-Group Differences

Significant differences were found between groups in the time duration component of the handwriting process, in favor of the neurotypical group (Z = 2.79; *p* = 0.005; control group M = 158.28, SD = 51.24, range 92.14–281.43; EFD group M = 217.17, SD = 109.45, range 88.18–506.00).

Significant between-group differences were noted in the HLS total score, with the control group presenting better legibility (Z = 3.34, *p* = 0.001, control group M = 8.15, SD = 1.88, range 5.00–12.00; EFD group M = 10, SD = 2.66, range 5.00–17.00). Table 1 presents these differences.

### 3.2. Correlations between Actual Handwriting Performance (Time and Legibility) and EF Subscale Scores in Each Group

No correlations were found between the control group’s time duration and BRIEF scale scores. However, in the EFD group, a significant negative correlation was found between time duration and the BRIEF shift scale (r = −0.36, *p* = 0.02).

In the control group, significant negative correlations were found between the HLS total score and the BRIEF parent reports of the shift scale (r =−0.55, *p* = 0.000), BRI (r = −0.32, *p* = 0.04), and working memory scale (r = −0.38, *p* = 0.02). Global legibility was negatively correlated with the shift (r = −0.54, *p* = 0.000) and working memory (r = −0.45, *p* = 0.003) scales. More negative correlations were found between the effort scale from the HLS and the shift scale (r = −0.46, *p* = 0.003), BRI (r = −0.39, *p* = 0.01), working memory scale (r = −0.41, *p* = 0.009), and GEC score (r = −0.34, *p* = 0.03). Layout on the page negatively correlated with the inhibition scale (r = −0.32, *p* = 0.05), shift scale (r = −0.57, *p* = 0.000), BRI (r = −0.39, *p* = 0.01), and GEC score (r = −0.32, *p* = 0.04).

In the EFD group, significant positive correlations were found between the alteration scale and the plan scale (r = 0.44, *p* = 0.004) and MI (r = 0.33, *p* = 0.04) and between the HLS total score and the plan scale (r = 0.39, *p* = 0.01).

### 3.3. Handwriting Anticipatory Awareness: Time Estimation

#### 3.3.1. Between-Group Differences

Significant differences were found between the groups in time estimation, with a higher time estimation in the EFD group (Z = 2.22, *p* = 0.03). In the control group, the mean time estimation was 11.08 min (SD = 5.91, range 3–30); the EFD group’s mean time estimation was 18.23 min (SD = 14.75, range 2–60).

#### 3.3.2. Differences between Time Estimation and Time Duration by Group

In the control group, significant differences were found between time estimation and time duration (Z = 5.40, *p* = 0.000). The time estimation mean was 348.00 s (SD = 166.52, range 180.00–900.00), whereas the actual time duration mean was 158.28 s (SD = 51.24, range 92.14–281.43). Of the 40 participants in the control group, 39 estimated more time than their actual duration, while the other estimated less.

Significant differences were also found between time estimation and time duration in the EFD group (Z = 4.76, *p* = 0.000). The time estimation mean was 491.70 s (SD = 282.77, range 120–960), and the time duration mean was 217.16 s (SD = 109.45, range 88.18–506). Of the 41 participants in the EFD group, 37 estimated more time than their actual duration, and the others estimated less.

#### 3.3.3. Correlation between the Time Estimation/Duration Gap and EF Subscale Scores by Group

In the control group, no significant correlations were found for the gap between time estimation and time duration with the BRIEF scales. However, in the EFD group, a significant positive correlation was found with the shift scale (r = 0.34, *p* = 0.03). The greater the gap, the more difficulty the participants had in shifting.

### 3.4. Handwriting Anticipatory Awareness: Expected Performance and Expected Difficulty

#### 3.4.1. Between-Group Differences

Table 2 shows the significant between-group differences found in expected performance (χ^2^ = 25.06, *p* < 0.000) and expected difficulty (χ^2^ = 20.12, *p* < 0.000) questions from the SAP-Q.

#### 3.4.2. Differences between Expected Performance and Expected Difficulty, and Legibility by Group

No chi-square correlations were found in the control group between the HLS total scores and the SAP-Q expected performance (χ^2^ = 0.70, *p* = 0.40) or the SAP-Q expected difficulty (χ^2^ = 0.70, *p* = 0.40). However, significant differences were found in the EFD group between the HLS total scores and expected performance (χ^2^ = 16.46, *p* = 0.002), but not expected difficulty (χ^2^ = 5.21, *p* = 0.26). Table 3 presents the distribution of expected performance and expected difficulty and the HLS total scores.

#### 3.4.3. Correlations between Expected Performance and Expected Difficulty, and EF by Group

In the control group, no significant correlations were found with the BRIEF. However, a significant correlation was found between expected difficulty and global legibility (r = −0.40, *p* = 0.01). In the EFD group, expected performance correlated with the inhibition scale (r = −0.42, *p* = 0.007), the HLS effort score (r = −0.40, *p* = 0.009), and the HLS total score (r = −0.40, *p* = 0.01).

## 4. Discussion

This study adds to the accumulated theoretical and clinical knowledge about the involvement of EF difficulties in adolescents’ daily activities. Their struggle highlights discrepancies with neurotypical peers in the handwriting activities required every day and the potential impact on their functional and academic achievement.

This study first aimed to identify significant differences between adolescents with and without deficient EF in their actual handwriting performance and anticipatory awareness and their correlation with EF components. The results showing differences between adolescents with and without EF difficulties in time duration support previous studies, including those that compared other daily tasks, such as cooking [43]. The results indicate that, compared with neurotypical adolescents, those with EFD need more time to produce texts [18,50].

Significant between-group differences were also found in the HLS total score, especially in effort required, letter formation, and layout on the page. Previous studies showed similar differences indicating that the EFD group’s product was less legible [23]. However, this study is the first to use the HLS with the EFD population. Previous results using the HLS showed significant differences between children with and without a developmental coordination disorder in the total score [49] and in each of its five component scores [46]. As such, this study also strengthens the HLS discriminant validity.

The correlations found in the EFD group highlight the plan component’s role in the handwriting product. More alterations (high HLS score) reflect low legibility, correlating with the difficulties in the BRIEF plan scale (high scores reflect more difficulties). The negative impact on legibility of alterations, including crossing out, adding strokes, and retracing letters to correct their form, has been identified [46,51]. Our results imply that the HLS alteration component depends on good planning skills to decrease alterations that impact product legibility.

The study’s second aim was to explore differences in how adolescents with and without EFD use anticipatory awareness, including their expected performance, difficulty, and time in the handwriting task, and the correlations with EF and their actual performance (time duration and legibility).

### 4.1. Handwriting Anticipatory Awareness: Time Estimation

The analysis of time estimation encompasses three viewpoints on the construct: between-group differences, within-group differences, and correlation with EF components.

The results reveal that the study group estimated the time it would take them to complete the handwriting copying task as longer than what the control group estimated. This assessment may be explained by their previous familiarity with their individual abilities—that completing handwriting tasks takes adolescents with EFD longer than it takes others. This estimation may indicate their self-awareness of their difficulty with completing handwriting tasks. Assuming that familiar tasks would be easier to perform and their processes would require less effort and stress, individuals with difficulties may be more likely to perform better with familiar activities [28].

However, when analyzing each group separately, the results indicated that an absolute majority of participants in both groups overestimated the time, stating that it would take them longer to complete the task than it actually did. Contrarily, most previous studies showed differences in the time estimations of children with and without difficulties and reported that adolescents with EFD underestimated their time. Thus, ours is an interesting and unexpected result.

Motivation to perform handwriting tasks might help explain this result [52,53]. Writing-task demands, including cognitive monitoring and appropriate strategies, require motivation to fuel the effort [54]. A concerning trend as students enter high school is their decreased motivation and value in writing by the time they finish middle school [55]. Gonida et al. [56] stated that children with negative illusory biases tended to report feeling less pride in their academic results overall along with less intrinsic motivation for mathematics and thinking more negatively about the effort they had to expend. These children found schoolwork more difficult and believed that doing it well required much effort. They set themselves less demanding achievement standards and adopted lower expectations of success. Adolescents familiar with handwriting tasks possibly had negative experiences and thus estimated their time based on their low motivation to do the task.

This analysis examined how the gap between the estimated time required to complete the handwriting task and the actual duration relates to EF components. The control group results showed no relationships with EF components, perhaps indicating that the overestimation stems from a lack of motivation to perform the task. However, among the adolescents in the EFD group, the results may stem partly from their EF difficulties, particularly with shifting.

The process of estimating time is complex, requiring an extensive range of other processes [57]. It includes understanding the duration and length of time, approximating how much time has passed within an activity, predicting the amount of time needed, and recognizing how long an activity has taken [58]. According to the dynamic attending theory (DAT), an “internal clock” is necessary to measure objective time subjectively without cues from “external clocks.” The DAT highlights the significance of the attention EF to the timing process overall [59]. It assumes that external environmental stimuli can influence the internal clock’s emission of temporal pulses [53]. When paying attention to external environmental cues (e.g., auditory or visual patterns), the internal clock’s speed may vary accordingly and distort the subjective perception of time (e.g., [60]).

Time perception studies have shown that children with ADHD are generally less accurate in time perception tasks [53,61,62]. They may likely shift their focus of attention continually in search of objects and events that are more stimulating in their surroundings. Thus, they might register more attentional pulses and perceive time as longer than it actually is. The shifting ability coordinates temporal and nontemporal processing, which is critical in virtually all interactions with the environment. Shifting is essential for coordinating and adjusting responses to the changing dynamics during environment–behavior interactions, including during sports or musical performances and while driving or handwriting [63].

As far as we determined, the literature is missing information about time estimation related directly to handwriting tasks. Considering the literature, we propose that the difficulties in the shifting component characterizing children with EFD, as assessed in this study, may hinder their ability to accurately estimate the time needed to complete the handwriting task. This is a crucial point from a clinical viewpoint: Therapeutic or educational work on the shifting component may help these children to more accurately estimate the time to complete tasks generally and handwriting tasks specifically.

### 4.2. Handwriting Anticipatory Awareness: Expected Performance and Expected Difficulty

We explored the expected performance and difficulty questionnaire for the time estimation using three analyses: between-group differences, the gap between the assessment and the actual performance (legibility), and correlations with EF.

The self-awareness between-group results indicate that most adolescents with EFD assessed their handwriting performance as bad or fairly good and expected difficulties (always to sometimes) in performing the handwriting copying task. According to Toglia and Kirk [27], feelings of effort and failure mainly influence subjective cognitive abilities. These beliefs may be obstacles to an individual developing healthy and adaptive self-awareness. In general, during the adolescence development period, children tend to judge their performance against others [64,65]. Adolescents with EFD struggle for years, particularly to bridge the gap between their impaired abilities and external environmental expectations [50], and harmful feedback that they receive may influence their self-awareness [28].

Similarly, teachers judge whether a student’s handwriting is quick and legible by comparing it with the student’s peers [66]. Regardless of content, papers with more precise and legible handwriting are more apt to receive higher grades than papers with poor penmanship [67]. Feedback to students can be positive or negative, direct or indirect [68]. Concrete feedback to adolescents with EFD may influence their handwriting self-assessments.

The control group of neurotypical teenagers showed more accurate anticipatory self-awareness for expected performance and task difficulty. Most estimated that they would perform the task very well and expected not to have to cope with difficulties during the task. Indeed, their handwriting was found to be at the excellent legibility level. As expected, there was no connection between their anticipatory assessment and actual performance, meaning there was no self-awareness impact on their product. Moreover, no correlations with the EF components were found.

Based on the literature, the EFD group would be expected to overestimate their expected performance [39,40]. However, this study’s results show otherwise. They demonstrate underestimation—anticipating low expected performance and more difficulties, although the product was excellent. Moreover, significant correlations were found between the expected performance and the inhibition component, which may help explain these results.

First, familiarity and low motivation for the handwriting task may be relevant here. Handwriting is a familiar, daily activity in which the participants were already involved. In grammar school, children often spend up to 50% of the school day writing assignments, some under time constraints [69]. Engaging in familiar, meaningful occupations can enable awareness to emerge because it provides a basis for comparing performance [70,71,72].

Students’ motivation for handwriting can fluctuate according to the task, content, and context [73,74], and their beliefs about themselves vary across writing tasks [75] and challenge levels [76]. Their motivation will shift over time according to their experiences [77]. Parents and teachers may view students who are unmotivated to write as having poor writing skills [78]. Even if this assessment is untrue, it might place the students on a path of specific coursework and lower expectations [55]. Children who have had negative experiences are apt to perceive high personal costs in engaging in the task and develop a poor attitude towards it [79]. Unfortunately, handwriting is challenging to master, and middle and high school students suffer from declining motivation.

Second, relationships between handwriting and inhibitory control and between decreased self-awareness and inhibitory control have been documented. However, to the best of our knowledge, there has been no specific reference regarded handwriting performance. Thus, preliminarily, we assume that the EFD group’s low inhibition ability hinders their ability to “stop and think” and analyze the task’s complexity systematically before estimating their expected performance [28]. If they did, perhaps their expected performance estimations would be more consistent with their actual performance.

Finally, in both groups, the significant correlations between expected performance (in the EFD group) and expected difficulty (in the control group) affect the product. Self-awareness affects performance. This fact indicates the need to evaluate self-awareness regarding performance in populations with writing difficulties, such as adolescents with EFD. Improving self-awareness before performing a challenging task such as writing might affect the product’s readability.

### 4.3. Limitations

This study has several limitations. Components other than EF that might influence handwriting as a skill, such as fine and gross motor skills and emotional components such as self-esteem, self-concept, and anxiety levels, are missing from these secondary data. Including them might have added insight into whether these data could separate the two levels of handwriting motor control. Further, despite the importance of the adolescents’ self-assessments, this study lacked parents’ and teachers’ input to complete the picture and help understand the gap. We recommend including parents’ and teachers’ input in future studies. Finally, anticipatory awareness is a less-studied concept in the literature; its structure and the correct and effective way to evaluate it are still not clear enough.

### 4.4. Conclusions

This study’s results strengthen EF as an underlying mechanism in the handwriting process. Product measures may be key for evaluation and intervention among those adolescents [18,80]. Moreover, it highlights anticipatory awareness of performance as an essential factor contributing to individuals’ ability to improve participation and functioning in daily activities [81]. Handwriting is an example of a daily activity that may inhibit the functioning of adolescents with EFD. These adolescents struggle to finish tasks that involve handwriting, copying from the board, writing in a reasonable time, or producing legible and spatially well-organized writing on the page. Moreover, they experience similar struggles in almost every day-to-day function they want and need to do. Unlike other populations, they underestimate their abilities regarding daily tasks and handwriting in particular.

This study’s results indicate the need to expand the research by examining different self-awareness constructs, including motor and emotional assessments, to explore why adolescents with EFD underestimate their handwriting performance. Future research should determine whether metacognitive intervention can help fill the gap between self-assessment and actual performance. We recommend that future studies examine existing self-awareness assessment tools and create uniform tools to compare adolescents with different difficulties and cultures and to better understand their functioning—as well as how to help them.

These results join other research evidence to emphasize the importance of adolescents’ self-assessment of their performance in daily activities such as handwriting. They could direct educators, parents, and therapists to help adolescents be aware of their performance, rate their difficulty, predict their performance, guide anticipation questions, and use strategy-generation techniques [28] before performing a handwriting task. Among adolescents with EFD, these conclusions may also direct the professional treatment process to emphasize the adolescent’s self-awareness before, during, and after a task, in order to develop effective strategies to perform the task and improve the performance. Moreover, after performing the handwriting task, positive feedback—including how the adolescents went about the activity and what they learned from the experience—will help build more accurate self-assessments for handwriting and other daily tasks.

## Figures and Tables

**Table 1 children-09-01628-t001:** Between-group differences in the HLS components.

HLS Component	M (SD)		Z	*p*
	Control (*n* = 40)	EFD (*n* = 41)		
Global legibility	1.43 (0.50)	1.66 (0.57)	1.81	0.070
Effort required	1.80 (0.46)	2.17 (0.67)	2.72	0.006
Layout on page	1.58 (0.59)	2.29 (0.78)	4.12	0.000
Letter formation	1.73 (0.45)	2.07 (0.65)	2.57	0.010
Alterations	1.63 (0.49)	1.80 (0.71)	1.01	0.310

Note: EFD, executive function deficits; HLS, Handwriting Legibility Scale.

**Table 2 children-09-01628-t002:** Between-group differences in process and product.

Expected Performance				Expected Difficulty			
Rating	Control (*n* = 40)	EFD (*n* = 41)	χ^2^	Rating	Control (*n* = 40)	EFD (*n* = 41)	χ^2^
Bad	0 (0)	3 (7.3%)	25.06 ***	Always	0 (0)	2 (4.9%)	20.12 ***
Fairly good	0 (0)	16 (39.0%)		Often	0 (0)	7 (17.1%)	
Good	10 (25%)	8 (19.5%)		Sometimes	10 (25.0%)	20 (48.8%)	
Very good	18 (45%)	8 (19.5%)		Seldom	21 (52.5%)	9 (22.0%)	
Excellent	12 (30%)	6 (14.6%)		Never	9 (22.5%)	3 (7.3%)	

*** *p* < 0.000.

**Table 3 children-09-01628-t003:** Crosstab test for distribution between the expected performance and expected difficulty, and the HLS total score.

HLS/SAP-Q		Control Group Legibility			EFD Group Legibility		
	Rating	Excellent	Good	Poor	Excellent	Good	Poor
Expected performance	bad + fairly good				6 (14.6)	13 (31.7)	
	good	10 (25)			6 (14.6)	1 (2.4)	1 (2.4)
	very good + excellent	28 (70)	2 (5)		12 (29.3)	2 (4.9)	
Expected difficulty	always + often				4 (9.8)	5(12.2)	
	sometimes	10 (25)			10 (24.4)	9 (22)	1 (2.4)
	seldom + never	28 (70)	2 (5)		10 (24.4)	2 (4.9)	

Values are presented as frequency (%).

## Data Availability

The datasets generated and/or analyzed during the current study are not publicly available due to ethical restrictions but are available from the corresponding author on reasonable request.

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
