# Peer review of "Anticipatory Awareness and Actual Handwriting Performance Measures among Adolescents with Deficient Executive Functions"

_children, 2022, doi:10.3390/children9111628_

Round 1
Reviewer 1 Report
This is a well-written manuscript. The paper reports findings from a secondary analysis of 81 adolescents and the relationships between temporal and qualitative features of handwriting, self-awareness, and clinical aspects of executive function deficits. The findings provide insight into self-awareness and graphomotor skill. Authors argue that while handwriting is an important skill necessary for academic achievement, handwriting performance metrics can serve to monitor clinical outcome and therapeutic intervention in children with ADD or EF deficits.
I have just a few comments and suggestions for the authors to consider.
1. Regarding the HLS ratings, an occupational therapist appeared to be the sole rater for all participants in this study. It is not clear that the rater was blind to EF diagnosis; nor how reliable his/her ratings were. Authors should elaborate on the subjective HLS ratings, especially with regard to reliability and managing bias.
2. Section 3.3.1 presents mean scores for time estimation. This was confusing as the methods described time estimation (line 167) as the subjective expectation of the participant. The scores in section 3.3.1 appear to be quantitative values derived from the digitizer. Perhaps the term "estimation" as used in 3.3.1 is not accurate. This should be clarified.
3. Lastly, the authors refer to both execution and production of handwriting in the Discussion (line 280). These seem to be the same concept, referring to the neuromotor component of handwriting. Little attention is given to the distinction between the programming (planning, sequencing) and execution of graphomotor behavior. This raises the question of whether handwriting abnormalities in adolescents with EF deficits (or ADD for that matter) stem from problems with higher level cognitive/programing (esp planning) or motor execution, either of which would lead to longer writing times or illegibility. Authors should address this and whether their data are capable of separating these two levels of handwriting motor control.
Reviewer 2 Report
Dear Authors
Very interesting theme, and relevance to research. Congratulations.
However, there are some issues to be reflected and, in my view, to improve.
I present my reflections according to your article.
Title
. “Handwriting Performance in Adolescents with Deficient Executive Functions: Do Actual Performance Measures Relate to Anticipatory Awareness?”- the title does not seem clear to me, does not report what was actually worked on in the study and does not reveal itself appealing.
Abstract
. EFD per parent reports- so we can't talk about executive function deficits (EFD) but rather parental perception of deficient executive functions. I believe that we should be very careful with this information, because there is no clinical diagnosis promoted by specialists.
. On the basis of what indicators have parents made this 'diagnosis'? It was through the Behavior Rating Inventory of Executive Function. It is essential to give this information in the summary.
. I also think it would be important for them to have applied this inventory to teachers in order to strengthen results. In my view, they should refer to deficient executive functions and not executive function deficits.
. Objective: you report that "estimated their time, performance, and difficulty before paragraph copying task using the Self-Awareness of Performance Questionnaire (SAP-Q)." How do you validate the goal? That is, time, performance and difficulty, is not clear in the summary. That is, to evaluate: Time, used ComPET; Performance, Self-Awareness of Performance Questionnaire (SAP-Q); and Difficulty-or-readability, HLS. It should be presented clearly. As well as the EFs and their dimensions that have been evaluated.
. You refer “Significant correlations were found between actual performance, anticipatory awareness, and EF in both groups”. They evaluated EF by parents. And why didn't they apply it to the teachers, too? It would be important from my perspective.
. You refer “The results strengthen EF role in the handwriting performance and highlight the relationships between anticipatory awareness and actual performance (time duration and legibility) of adolescents with and without EFD.” It is not clear these results are presented as evidence, but in the abstract we have no information on how they were measured.
. You put as keywords: adolescent, handwriting, executive function deficit, self-awareness, time estimation, copying, Handwriting Legibility Scale. The keywords in my perspective do not represent the study.
1. Introduction
. Line 82- You refer “adolescents with deficient EF”- evaluated how?
. Line 86- ADHD- You haven't told them what it is yet and put an acronym (Attention Deficit Hyperactivity Disorder). Should be reviewed.
. Lines 101-103- You refer “However, there is a lack of evidence on how adolescents’ self-awareness before performing a handwriting task (anticipatory self-assessment) can affect their actual handwriting-performance process (time duration) and product (legibility).” But is performance evaluated by time? Isn't that SAP-Q? Confused. It should be clarified.
. Lines 104-107- You refer “Thus, this study’s aims were to (1) compare actual handwriting performance (time duration and legibility) and anticipatory awareness of adolescents with and without deficient EF and (2) analyze correlations between actual performance, anticipatory awareness, and EF components in each group.” They are not in line with the objective they present in the abstract. It should be reviewed and translated into the results.
2. Materials and Methods
2.1. Participants
. As exclusion criteria you do not approach the taking of psychotropic drugs. As literature indicates, these influence EFs. Was it taken into consideration? If they should not put as limitation of the study.
. I think that an emotional adjustment screening should have been carried out. They only report that “Participants with known psychiatric, emotional, or autism spectrum disorders, physical disabilities, or neurological diseases were excluded.” Because the fact that they don't have disorders doesn't mean they're emotionally adjusted.
2.2. Procedure
. The procedure of the study is not made known, only share the ethical issues. I think you should get better.
2.3. Instruments
2.3.2. Handwriting Process: Computerized Penmanship Evaluation Tool (ComPET
. Lines 144 and 145- You refer “for the current analysis, time duration (in s) data were extracted from the ComPET”. Why this selection? It should be explained.
3. Results
. In my opinion, the results should be written on the basis of objectives that should have been clearly presented earlier. They are not in line with the provisions of the lines 104-107, where you refer “Thus, this study’s aims were to (1) compare actual handwriting performance (time duration and legibility) and anticipatory awareness of adolescents with and without deficient EF and (2) analyze correlations between actual performance, anticipatory awareness, and EF components in each group”. They are not in line with the objective they present in the abstract. It should be reviewed.
4.3 Limitations
. I agree, however, they should indicate which cognitive and emotional abilities are most evidenced by the literature and which in the future should be screened/evaluated. In my view, being a study that evaluates Anticipatory Awareness was essential to perform a misdemeanor at the level of self-esteem, self-concept and anxiety.
4.4 Conclusions
. It could be further explored and identify contributions to comprehensive analysis and practical intervention.
